# Essential-Oils-Loaded Biopolymeric Nanoparticles as Strategies for Microbial and Biofilm Control: A Current Status

**DOI:** 10.3390/ijms25010082

**Published:** 2023-12-20

**Authors:** Alejandra Romero-Montero, Luis Javier Melgoza-Ramírez, Jesús Augusto Ruíz-Aguirre, Alejandra Chávez-Santoscoy, Jonathan Javier Magaña, Hernán Cortés, Gerardo Leyva-Gómez, María Luisa Del Prado-Audelo

**Affiliations:** 1Departamento de Farmacia, Facultad de Química, Universidad Nacional Autónoma de México, Mexico City 04510, Mexico; alejandra.romero.montero@outlook.com (A.R.-M.); leyva@quimica.unam.mx (G.L.-G.); 2Tecnologico de Monterrey, Escuela de Ingeniería y Ciencias, Mexico City 14380, Mexico; luisjavmr@tec.mx (L.J.M.-R.); a01658888@tec.mx (J.A.R.-A.); magana.jj@tec.mx (J.J.M.); 3Tecnologico de Monterrey, Escuela de Ingeniería y Ciencias, Campus Monterrey, Ave. Eugenio Garza Sada 2501 Sur, Monterrey 64849, Mexico; chavez.santoscoy@tec.mx; 4Laboratorio de Medicina Genómica, Departamento de Genómica, Instituto Nacional de Rehabilitación Luis Guillermo Ibarra Ibarra, Mexico City 14389, Mexico; hcortes@inr.gob.mx

**Keywords:** biofilm, essential oils, chitosan nanoparticles, phytochemicals, natural products

## Abstract

The emergence of bacterial strains displaying resistance to the currently available antibiotics is a critical global concern. These resilient bacteria can form biofilms that play a pivotal role in the failure of bacterial infection treatments as antibiotics struggle to penetrate all biofilm regions. Consequently, eradicating bacteria residing within biofilms becomes considerably more challenging than their planktonic counterparts, leading to persistent and chronic infections. Among various approaches explored, essential oils loaded in nanoparticles based on biopolymers have emerged, promising strategies that enhance bioavailability and biological activities, minimize side effects, and control release through regulated pharmacokinetics. Different available reviews analyze nanosystems and essential oils; however, usually, their main goal is the analysis of their antimicrobial properties, and progress in biofilm combat is rarely discussed, or it is not the primary objective. This review aims to provide a global vision of biofilm conformation and describes mechanisms of action attributed to each EO. Furthermore, we present a comprehensive overview of the latest developments in biopolymeric nanoparticles research, especially in chitosan- and zein-based nanosystems, targeting multidrug-resistant bacteria in both their sessile and biofilm forms, which will help to design precise strategies for combating biofilms.

## 1. Introduction

Bacterial resistance to antibiotics, especially in biofilms on medical devices or tissues, is a global health concern due to the emergence of strains resistant to virtually all available antibiotics [1]. Biofilms, slimy layers formed by bacteria on implanted devices or damaged tissue, enhance antibiotic resistance by providing a protective matrix [1]. Biofilm bacteria are 1000 times more antibiotic-resistant than individual cells, making infections chronic and difficult to treat [1].

Resistance mechanisms in biofilms differ from individual bacterial cells, relying on multicellular strategies and hindering antibiotic penetration into the biofilm [2]. Microbial antibiotic resistance is a significant global health issue, ranking among the top eight until 2020 [3]. Biofilm-related infections in medical settings have a higher treatment failure incidence, posing life-threatening risks and necessitating device or tissue removal [4].

To address this health challenge, interdisciplinary efforts focus on developing drugs and materials for localized antibiotic administration. Essential oils (EOs) like cinnamon, rosemary, and clove oils, rich in terpenes, phenols, and aldehydes, exhibit antimicrobial, antifungal, and anti-inflammatory properties [5,6]. However, these phytochemicals face challenges such as limited stability, solubility, and volatility, impacting their therapeutic use and risking allergic reactions [7,8].

Nanotechnology offers a solution to enhance the bioavailability and biological activity of bioactive compounds, control their release, and minimize side effects, thereby improving therapeutic outcomes [9,10,11]. Biopolymer-based nanocarriers, particularly those utilizing chitosan and zein, show promise for encapsulating essential oils (EOs) due to their biocompatibility, low toxicity, and surface modification potential [8,12,13,14,15].

These natural biopolymers act as convenient carriers for drugs and active molecules, possessing biocompatibility, biodegradability, and antibacterial activity. Polysaccharide-based nanoparticles have gained interest in nanomedicine, biomedical engineering, and therapeutic delivery systems [15].

While various nanosystems have been studied for controlled release of EOs with antibacterial activity [16,17,18], research on the antibiofilm activity of biopolymeric nanocarriers loaded with EOs or EO-derived phytochemicals is limited. The existing literature mainly focuses on the antimicrobial properties of these systems, with scant discussion on their progress in combating biofilms [7].

This review aims to offer an extensive overview of the latest advancements in chitosan and zein nanoparticles (NPs) research, specifically targeting multidrug-resistant bacteria in both their sessile and biofilm forms. The selection of these biopolymers is based on their potential applications in biomedical fields. Furthermore, a comprehensive analysis of alternative treatments against antibiotic-resistant bacteria and biofilm persistence is provided by evaluating various other biopolymeric nanocarriers. This broad exploration underscores the relevance of nanoparticles in addressing the challenges posed by antibiotic-resistant bacteria and persistent biofilms. This approach not only enhances our understanding of the inhibitory phenomenon but also facilitates the development of precise strategies for combating biofilms.

## 2. Microbial Biofilm

Bacteria utilize biofilm formation as a survival strategy against external threats, including the host’s immune system, antibiotics, and environmental factors. Biofilms, clusters of bacteria adhering to surfaces and each other, are encased in a self-produced matrix comprising proteins, polysaccharides, and DNA, providing protection to the bacterial colony [19]. This matrix enables the colony to remain inactive and concealed from the immune system [20].

Within the biofilm, bacteria adapt to oxygen and nutrient deprivations by altering their metabolism, limiting gene expression, and reducing protein production. This adaptive response slows down metabolism and decreases the rate of cell division [21].

### 2.1. Biofilm Structure 

Biofilms, intricate three-dimensional structures, are composed of single cells or complex communities that produce extracellular polymeric substances (EPS) forming the matrix. This matrix includes DNA, RNA, structural proteins, lipids, enzymes, and polysaccharides. Humic and uronic acids are also common in biofilms [22]. The extracellular matrix, constituting 40–95% of EPS, offers robust protection against external threats, with microorganisms occupying less than 35% of the biofilm volume [23].

Within the biofilm, bacterial assembly in layers connected by matrices and channels enables nutrient, water, and air exchange. Microbial cells in biofilms exhibit altered phenotype, genetic regulations, and increased antimicrobial resistance compared to planktonic counterparts [24]. Biofilm shape and distribution are influenced by factors like surface properties, initial cell distribution, bacterial strain, gene transcription, quorum sensing, and environmental physicochemical properties [21].

While most biofilms are flat aggregates, some, like those formed by *Pseudomonas aeruginosa*, have mushroom-like shapes [25]. The extracellular matrix restricts nutrient transfer, leading to diverse microenvironments and heterogeneous cell distributions. Diffusion is the primary nutrient transfer mechanism within biofilms, affecting the metabolism and genetic transcription of different populations. Social behavior among bacteria enables organization and survival, a topic of growing interest among microbiologists [26,27,28].

### 2.2. Formation Stages 

The biofilm formation process encompasses a complex intercellular and intracellular signaling cascade, typically described in five stages (Figure 1). Initially, bacteria approach a biotic or an abiotic surface, reduce mobility, and adhere through Van Der Waals forces. Subsequently, a monolayer forms, and bacteria produce EPS, serving as a cohesive glue that binds them together and to the surface [23,29].

In the third stage, bacteria multiply, divide, and create microcolonies, contributing to the three-dimensional structure of the film. The formation of layers and small cell clusters follows, leading to the building and maturation of the biofilm structure [30,31]. Microcolonies create channels as bacteria move, facilitating oxygen, nutrient, and water supply to all biofilm cells. In the final stage, chemical signaling prompts the overexpression of genes involved in flagella formation, allowing bacteria to exit the biofilm and colonize other surfaces [32,33].

The survival strategy involves rapid multiplication, detachment, dispersion, and colonization, covering a larger surface area and making infection control challenging [34,35]. The transition to motile bacteria may address the need for movement and colonization. Intracellular signaling, triggered by nutrient and oxygen scarcity and an abundance of waste molecules, induces stress, prompting a phenotype modification to “escape” the biofilm and initiate a new colony [20,23].

### 2.3. Biofilm Resistance to Antibiotics Mechanisms 

Biofilm formation exacerbates the ongoing global health challenge of antibiotic resistance, reducing antibiotic sensitivity by approximately 100-fold in this bacterial organization. Several resistance mechanisms exist in biofilm-organized bacteria, with some notable examples.

One resistance mechanism involves the physicochemical interaction of antibiotics with the extracellular polymeric substances (EPS) in the biofilm. The EPS matrix acts as a physical barrier, hindering antibiotic diffusion through the layers. Antibiotics can be trapped through chelation reactions or degraded by enzymatic reactions within the matrix. For instance, alginates in the biofilm of *P. aeruginosa* can block the action of gentamicin or aminoglycosides by forming bonds with cationic antibiotics [36].

Stress conditions within the bacterial community provide another protective mechanism. Nutrient scarcity induces some bacteria to enter a dormant phase, activating biological defense mechanisms such as thickening the cell wall. This results in antibiotics targeting only metabolically active cells, while dormant cells can survive until conditions improve [37,38].

The collective strength of a biofilm also contributes to resistance; a higher number of cells enhance resistance. While a single cell may produce an enzyme or metabolite to counteract antibiotics, enough cells can collectively fight the toxic effects. Additionally, diverse strains with different abilities can coexist to protect the community. For example, catalase-producing cells shield non-producing cells from H_2_O_2_, and aerobic cells protect anaerobic cells from oxygen. Prolonged exposure to adverse conditions and defense mechanisms leads to gene expression modifications, resulting in a generation of cells naturally resistant to antibiotics [34,39].

## 3. Essential Oils with Antibiofilm Properties 

Essential oils (EOs), including cinnamon, rosemary, clove, and oregano oils, have garnered significant attention from the scientific community. Comprising primarily aldehydes, terpenes, and phenols, these EOs exhibit diverse biological activities, such as antimicrobial, antifungal, antiprotozoal, antioxidant, anti-inflammatory, and anticancer properties [40,41,42,43].

The use of natural oils, as opposed to isolating pure components, enhances the synergistic action of phytochemicals, enabling them to act through various steps to boost antimicrobial effectiveness. For instance, lipophilic compounds like terpenes and phenols or those with functional groups can target cell membranes [44,45] (Figure 2a). Compounds like cinnamaldehyde and eugenol can interact with proteins and enzymes (Figure 2b), allowing them to enter the cytoplasm and mitochondria, thereby inhibiting protein and ATP production necessary for bacterial survival [46,47,48] (Figure 2c). Moreover, components of natural oils have been identified to disrupt the mechanism of efflux pumps [49,50] (Figure 2d), which are developed by strains exhibiting multidrug antibiotic resistance.

### 3.1. Cinnamon Oil 

Cinnamon has gained widespread use in global traditional medicine [51]. Renowned for its antibacterial, antifungal, antioxidant, anti-inflammatory, insecticidal, anticancer, pesticide, and antidiabetic properties, both its bark and leaves contribute to its diverse benefits [52,53].

Natural cinnamon oils typically include trans-cinnamaldehyde (TC), cinnamyl acetate, eugenol, L-borneol, camphor, caryophyllene oxide, b-caryophyllene, L-bornyl acetate, E-nerolidol, α-cubebene, α-terpineol, terpinolene, and α-thujene [54]. Despite cinnamon’s broad health benefits, Gram-negative bacteria exhibit higher resistance to the bactericidal effects of natural oils, including cinnamon essential oils (CEO), primarily due to porin proteins that permit the passage of hydrophilic components while blocking hydrophobic ones [55].

One of the key components of cinnamon essential oil (CEO) is trans-cinnamaldehyde (TC). This phenylpropene could penetrate the bacterial cell wall’s phospholipid bilayer, binding to proteins and disrupting membrane permeability. It further blocks the transport of proteins and ions, inducing bacterial death through cytoplasmic coagulation, denaturation of proteins, and loss in metabolites and ions. TC also imparts rigidity to the bacterial membrane by saturating a significant proportion of fatty acids in the phospholipid membrane [56].

Beyond TC, CEO contains other components that inhibit the electron transport chain in mitochondria, hindering ATP generation [57]. Additionally, it inhibits the GTP-dependent polymerization of protein FtsZ, a regulator of bacterial cell division [58]. CEO influences the expression of bcsA and luxR, key players in quorum sensing, the cell communication system essential for organizing the biofilm structure [59].

### 3.2. Rosemary Oil

Rosemary (*Rosmarinus officinalis* L.), a shrub in the Lamiaceae family, is renowned for its organoleptic qualities and robust antimicrobial and antioxidant activities [60]. With a rich history in traditional medicine, it has been used as a pulmonary antiseptic, choleretic, cholagogue, antidiarrheal, antirheumatic, and, more recently, recognized for its antiviral, anti-inflammatory, and anticancer properties [61].

The chemical composition of rosemary essential oils (RMEO) includes eucalyptol, α-pinene, β-pinene, *p*-cymene, camphor, linalool, and γ-terpinene [62,63]. Despite multiple reports on RMEO’s antimicrobial activity, a consensus on its mechanism of action is elusive. Effective against Gram-positive strains, RMEO demonstrates minimum inhibitory concentrations (MIC) ranging from 0.020 µg/mL to 0.8 µg/mL [64,65,66]. Some studies attribute its antimicrobial properties to the high concentration of terpenes, causing cell membrane disorganization and lysis [67,68]. 

Others link bactericidal activity to polyphenols, despite their minority presence [69]. Polyphenols interact with cell membrane proteins, altering structure and functionality, disrupting mitochondrial electron transport, and causing damage to genetic material and essential nutrients [70]. Fatty acid production may also be affected [71]. Some authors suggest RMEO’s effectiveness results from the synergy between polyphenols and terpenes, regardless of their proportions [61].

### 3.3. Clove Oil

Clove essential oil (ClEO) (Syzygium aromaticum) primarily comprises over 80% eugenol, alongside smaller proportions of eugenol acetate, β-caryophyllene, and other trace components [72,73]. ClEO exhibits a versatile range of properties, including antibacterial, antioxidant, insecticidal, antifungal, anti-inflammatory, antiviral, anticancer, analgesic, antispasmodic, and organoleptic effects [74,75].

The antibacterial properties of ClEO, closely linked to eugenol, involve a mechanism of action like trans-cinnamaldehyde (TC) on Gram-negative strains. Eugenol’s chemical properties enable it to cross the lipopolysaccharide membrane and enter the cytoplasm, resulting in the loss in cellular components [47]. The presence of a free -OH group in ClEO’s structure contributes to its antibacterial activity on Gram-positive strains like Pseudomonas aeruginosa. The -OH group of eugenol is believed to inhibit certain enzymes and generate reactive oxygen species (ROS) intracellularly [76], leading to bacterial cell death by inhibiting growth, disrupting cell membranes, and damaging DNA [48].

ClEO inhibits biofilm formation, reducing biomass production by up to 50% in vitro and in vivo on *S. aureus* biofilms. Eugenol interferes with the expression of biofilm-associated genes such as IcaD (intracellular adhesion gene) [77] and inhibits quorum-related virulence factors [78].

### 3.4. Oregano Oil

Oregano essential oil (OEO) has a history of traditional medicinal use for treating asthma, diarrhea, indigestion, contraception, and cramps due to its organoleptic properties [79,80]. Recent studies highlight OEO’s potent antioxidant and antibacterial effects, primarily attributed to constituents like carvacrol, thymol, trans-caryophyllene, α-humulene, o-cymene, and γ-terpinene [81].

While the precise antimicrobial mechanism of OEO is not fully elucidated, carvacrol, a phenolic compound, is believed to play a significant role. Carvacrol induces functional and structural alterations in the cell membrane, disrupting nucleic acid synthesis and functionality, expelling intracellular components, causing metabolic imbalance, and interfering with quorum sensing to prevent biofilm spread and formation [82,83].

Carvacrol’s ability to cross the lipid bilayer destabilizes it, leading to membrane expansion and increased permeability. This triggers a drop in internal pH and a loss in the electrochemical membrane potential, impacting ATP synthesis. Experimental evidence suggests that the efflux of essential ions, such as H^+^ and K^+^, is crucial for microbial cell death. Carvacrol’s -OH group acts as an ion exchanger, promoting cation exchange in bacterial cells. Upon interaction and crossing the membrane, carvacrol reaches the cytoplasm, losing the acid proton in its uncharged form [84].

Exposure to carvacrol induces elevated levels of heat shock proteins and non-motile cells in *E. coli* O157:H7, causing physiological and morphological alterations linked to variations in peptidoglycan composition and inhibiting biofilm formation [84]. OEO further interferes with quorum sensing by carvacrol’s interaction with ExpI/ExpR proteins, depleting N-acyl homoserine lactone, a signal molecule crucial to produce plant-cell-wall-degrading enzymes [85].

## 4. Nanoparticles’ Characteristics as an Antibiofilm Strategy 

### 4.1. The Effect of Size and Shape

Biofilms can have different components, including polysaccharides, proteins, lipids, and extracellular DNA, depending on the strain that produces them. Biofilms formed by more than one kind of microorganism can have more complex compositions, making it harder to predict how NPs will interact with them. The interaction of NPs with biofilms is described in a two-step sequence: attachment to the biofilm surface and migration within the biofilm. The effect of NPs in a biological system depends on various factors, such as composition, size, shape, roughness, surface properties, and ability not to form aggregates. All these parameters are related, and multiple applications may lead to an apparent contradiction between the authors. Therefore, it is not easy to draw conclusions regarding an isolated parameter.

Concerning particle size, there is some evidence that a decrease in size can increase the inhibition capacity of microorganisms [86]. This trend can continue in a specific range of particle size reduction until reaching a point where the biological effect does not change. Sikder et al. demonstrated the impact of the size in micellar systems on *Staphylococcus aureus*, finding that smaller nanomaterials presented a higher binding constant in assays with lipid membranes [87]. Raghupathi et al. determined that the antibacterial activity of NPs (via the production of ROS) in *Staphylococcus aureus* was inversely proportional to the size (assessed from 12 to 307 nm), associated with an MIC of 4 to 7 mM. The study found a critical biological effect in a size range larger than 100 nm, indicating an interaction phenomenon dependent on the microorganism strain [88]. Also, it is essential to note that a decrease in NP size increases the area of exposure and the surface free energy, resulting in a higher tendency towards flocculation and coalescence, which may reduce the effect against microorganisms. Therefore, the impact of the surfactant and the Zeta potential will be crucial. Interestingly, some resistance mechanisms of microorganisms include the expression of genes that favor NP flocculation to decrease cytotoxic activity, a strategy that cannot be resolved by using surfactants [89].

On the other hand, reducing the size of NPs can significantly increase damage to surrounding cells, such as fibroblasts. A study demonstrated that decreasing the size of silver NPs led to increased damage to L929 fibroblasts due to enhanced ROS production, which is associated with a larger surface area [90]. Therefore, decreasing the size of NPs can result in a bactericidal effect, while a gradual increase in size can induce a bacteriostatic effect until it loses its effectiveness against microorganisms. Additionally, the extent of inhibition will depend on the type of microorganism and the concentration of NPs [88]. 

Additionally, in the NP–biofilm interaction, the diffusion coefficient decreases exponentially with the square of the NP’s radius [91]. In this regard, Peulen and Wilkinson elucidated in a study on biofilm composed of Pseudomonas fluorescens (by fluorescence correlation spectroscopy) that the self-diffusion coefficient also decreases for a dense bacterial biofilm [92]. Also, the study evidenced the influence of the crosslinking of the polymer chains and the pore size during biofilm penetration. The greater the interaction of the polymer chains and the smaller the pore size, the lower the diffusivity of NPs. Additionally, the authors found that the adequate pore size was 50 nm for the free flocs and 10 nm for the dense biofilms, indicating that larger NPs in a dense biofilm have low diffusion and penetration [92]. Therefore, an increase in the diffusion coefficient with an increase in temperature and a decrease in the diffusion coefficient with an increase in the viscosity of the biofilm microenvironment could be expected.

Concerning the shape of the NPs, some studies suggest that the rod-like shape is more effective than the spherical particles. Such is the case in the study by Slomber et al., who reported that NPs changed the aspect ratio against *Pseudomonas aeruginosa* and *Staphylococcus aureus* biofilms [93]. These results agree with another study that used silica nanorods to release nitric oxide against Gram-positive *Staphylococcus aureus* and Gram-negative *Pseudomonas aeruginosa* bacteria [94]. Likewise, different cell lineages have shown that rod-like NPs increase mammalian cell internalization with low cytotoxicity and circulation time compared to their spherical counterparts [95,96]. Therefore, higher aspect ratios are more effective than the sphericity of NPs (Figure 3).

### 4.2. Functionalization

The high exposure area of the NPs favors the interaction with macromolecules through adsorption phenomena to form different interaction layers on the surface. The natural organic matter corona comprises the adsorption of environmental analytes, and the biomolecular corona includes the adsorption of biofilm components [91]. The constitution of the corona will increase the radius of the NPs–extracellular polymeric substances complex, with a possible decrease in the diffusion coefficient. Furthermore, forming a surface layer can decrease the drug release rate. Therefore, functionalization processes can increase the penetration capacity and the effect against bacteria. Functionalization processes can start with the role played by the electrical charge on the surface. Positively charged NPs interact more with the biofilm’s EPS, such as polysaccharides, proteins, and DNA.

Moreover, hydrophobic NPs can better colocalize with bacteria, while hydrophilic ones only with EPS. Therefore, it is possible to search for a synergistic effect of electric charge and the presence of functional groups [91]. Other systems may include enzyme-functionalized NPs or enzyme mimicry, for example, decoration with proteinase K for disrupting Pseudomonas fluorescens biofilms’ structure or using DNase-like artificial enzyme for cleavage of extracellular DNA of biofilm [96,97]. Combining pH-sensitive polymers can be a convenient alternative to take advantage of the acid environment inside the biofilm. The change from a neutral pH to the acid medium increases the relaxation of the polymer chains of the NP’s core with the subsequent release of a bactericidal substance [98]. 

### 4.3. Controlled Release

Typical NPs used for penetrating biofilms may contain antimicrobials, photosensitizers, or enzymes [98]. It is crucial to ensure the complete delivery of the cargo to maintain the MIC and prevent its release in other areas that could promote antibiotic resistance. In nanosphere and nanocapsule systems, it is also essential to achieve a gradual release over time while reaching the MIC within a short period. Otherwise, although the NP may possess high penetration capacity within the biofilm, a bactericidal or bacteriostatic effect might not be observed.

## 5. Recent Progress of Essential Oils Entrapped in Biopolymeric Nanoparticles as Antimicrobial Strategy

EOs have many applications given their various benefits, such as antibacterial, antifungal, anti-inflammation, and antioxidant properties [99]. However, their physicochemical properties (low solubility in water, volatile nature, and instability under high heat, light, and oxygen) make EOs unsuitable for several applications [100]. Delivery systems based on macromolecules such as chitosan or alginate represent an alternative to entrap EOs since these approaches improve the solubility of hydrophobic compounds, biodistribution, bioavailability, and drug targeting and limit their degradation and toxicity [101,102]. This section discusses the most relevant studies on EOs entrapped in different macromolecule-based NPs, such as chitosan, zein, and other biopolymers, and their current status (Table 1).

### 5.1. Chitosan-Based Nanoparticles

Chitosan is derived from the deacetylation process of chitin, which is a linear amino polysaccharide composed of N-acetylglucosamine units connected by β-(1,4) bonds. The degree of deacetylation of chitosan exceeds 50%, and it readily dissolves in weak acids (pH < 6) [119]. Naturally, as a biopolymer, its chain size and various physicochemical characteristics are influenced by factors such as the source (insect and mollusk shells) and the extraction method. From the chemical point of view, the presence of amino groups provides unique attributes by protonating NH^3+^ and increasing its solubility. This characteristic allows the formation of nanostructures through intermolecular interactions and presents biological functions, such as antimicrobial activity [120]. Primary amine groups also exhibit high reactivity upon nucleophilic addition to aldehydes, followed by dehydration to form imine bonds (Schiff bases). This chemical property allows functionalization with molecules that improve their mechanical and structural properties [121]. Chitosan is widely used to encapsulate EOs due to its ability to form gels, films, and particles and its biodegradability, non-toxic, and biocompatibility properties [105,122].

Other materials like liposomes as nanocarriers of EOs have also been studied; however, these carriers present problems such as instability, low drug loading capacity, and rapid release of entrapped drugs [123]. Several published studies have reported NP release systems of EOs with biopolymers [124], each employing different approaches and formulations. For example, Hadidi et al. [103] encapsulated clove ClEO in chitosan NPs using a two-step technique of emulsion–ionic gelation. The authors obtained a system with particle sizes between 223.2 and 444.5 nm. CEO-loaded chitosan NPs’ in vitro inhibitory activity was tested with *S. Aureus*, *L. monocytogenes*, *S. typhi*, and *E. coli*, demonstrating inhibition halo values of 4.8, 4.78, 4.49, and 3.95 cm, respectively. 

Similarly, Hasheminejad et al. [104] encapsulated ClEO in chitosan NPs to assess the antifungal enhancement of ClEO. The formulation presented a size range of 40 to 100 nm, suggesting that the ionic and anionic solution ratios can impact particle size [125]. The effect of ClEO NP in inhibiting *A. niger* growth was significantly higher than that of free ClEO. The results of both studies suggest that encapsulation prevents EOs evaporation and controls the release of the preserved volatile oil from chitosan NPs [126], improving its components’ antioxidant and antibacterial activity compared to their free forms [127].

Barrera et al. [121] extracted cinnamon, thyme, and Schinus molle EOs, and these were encapsulated by ionotropic gelation in chitosan NPs to assess the antimicrobial activity compared with the Eos’ free forms. According to the results, CEO-loaded NPs exhibited a spherical shape and a size of 29.3 ± 0.9 nm. The in vitro antimicrobial assay demonstrated that CEO NPs inhibited the proliferation of *K. pneumoniae* and *Enterococcus* sp. to 20.1 and 13.7%, respectively, in contrast to their free form, with a value of 75.2 and 39.7%, respectively. Similarly, Hu et al. [105] formulated chitosan NPs loaded with CEO by ionic gelation, obtaining a particle size range of 112–527 nm. The antimicrobial assay was performed in samples of chilled pork, one as a control and the other treated with the CEO NPs. The formulation extended the storage time from 6 to 15 days compared with the control. The results of both studies suggested that the nanoencapsulation of CEO enhances the antimicrobial activity and controlled release in contrast to the free form.

### 5.2. Zein-Based Nanoparticles

Zein is a corn prolamin, rich in hydrophobic amino acids such as proline, leucine, glutamine, and alanine, which make it soluble in ethanol at concentrations over 70% [111,128]. Hydrophobicity makes zein capable of self-association into NPs in polar solvents such as water. In recent years, zein NPs have attracted attention in the biomedical field for encapsulating natural molecules [129].

For example, in 2021, Gong et al. encapsulated thymol in zein NPs in different ratios between thymol and zein to evaluate their effect on *E. coli* and *S. aureus* [130]. The NPs with an average size from 119.1 to 167 nm allowed entrapped percentages from 45 to 60% of the EO. The best formulation, based on the encapsulation efficiency (thymol:zein, 3:5), was evaluated by disc-diffusion assay against the strains showing a higher inhibition zone than the unencapsulated thymol, demonstrating the improvement of the thymol properties in nanosystems.

Similarly, *E. coli*, *S. aureus*, and *L. monocytogenes* were employed to analyze the effect of zein NPs loaded with EOs of *Thymbra capitata* (L.) Cav. The authors also evaluated the formulation as a suspension to find an adequate system regarding bacteriostatic and bactericidal effectiveness [131]. Their findings suggested that both systems were active against the growth of Gram-positive and Gram-negative strains; however, isolated colonies grew randomly inside the inhibitory halos, but the NPs presented a higher activity than the suspension (4-fold higher in *E. coli* and 7-fold higher in *L. monocytogenes*), making them a more efficient system.

As mentioned above, the functionalization of NPs could change the interaction of the nanocarriers with the target. Carvacrol-loaded zein NPs were modified on the surface using polydopamine to enhance their effect on *S. typhimurium*, *P. aeruginosa*, and *S. aureus* [132]. The authors reported that the antibacterial activity of NPs without functionalization was around 70% due to the high properties of carvacrol; however, the modified NPs presented 100% inhibition for the three strains with the lowest concentration of polydopamine. These results demonstrated the potential of macromolecule-based NPs as new alternatives to combat infections, using natural molecules and exploiting the advantages of the surface and size.

### 5.3. Alginate-Based Nanoparticles

Alginate is a polysaccharide extracted from seaweed that is characterized by containing α-l-guluronic acid and β-D-mannonic acid residues, which are linearly linked by a 1,4-glycosidic bond. The carboxylate groups within the G units confer a global negative charge at pH = 7, typically compensated by utilizing Na^+^ cations as counterions [133]. The chemical structure o alginate contains multiple reactive groups, which make it highly flexible and suitable for chemical modifications [134]. These modifications are facilitated by carboxyl and hydroxyl groups within alginate, enabling noncovalent bond formation. As a result, alginate exhibits enhanced bioadhesion to mucous membranes, improving the bioavailability of problematic drugs. Under mild conditions, the coupling of alginate with tri- or divalent cations allows for the effective entrapment of drugs [135].

In 2021, Almasi et al. [136] encapsulated thyme EO within alginate, resulting in NPs with a size of approximately 100 nm. The antibacterial activity of these alginate-encapsulated thyme oil formulations was investigated using the diffusion assay method against strains of *E. coli* and *S. aureus*. The results showed the formation of an inhibition halo measuring approximately 8.5 mm, with no significant differences observed between Gram-positive and Gram-negative bacteria. Furthermore, when these NPs were applied to refrigerated meat, they exhibited a reduction of approximately 10% in the colony-forming units (CFU) of aerobic mesophilic microorganisms. These microorganisms represent viable organisms in food, including coliforms, *S. aureus*, lactic acid bacteria (LAB), molds, and yeasts. The reduction in CFU was observed after eight days of storage, indicating the potential antimicrobial efficacy of the alginate-encapsulated thyme oil NPs in preserving the quality and safety of refrigerated meat [136].

In a recent study by Santos et al. [137], alginate NPs were developed to encapsulate EOs of tea tree, cucumber, and a combination of both. This was achieved using the emulsification method followed by crosslinking with CaCl_2_. The resulting NPs had a size ranging from 200 to 400 nm, a Zeta potential between −15 and −40 mV, and an encapsulation efficiency of approximately 85%. The authors employed the diffusion assay method to assess the antimicrobial activity of these formulations, focusing on *E. coli* and *S. aureus*. The results revealed that the NPs loaded with tea tree oil exhibited a significantly larger inhibition halo, demonstrating their potent antimicrobial effects. Interestingly, the addition of cucumber seed oil did not enhance this capacity. Notably, EOs typically exhibit more potent effects against Gram-positive bacteria, which aligns with the observed results in this study.

In 2021, Yoncheva et al. [16] formulated OEO NP of chitosan and alginate by emulsification and consequent electrostatic gelation of chitosan and alginate. The formulation was analyzed in different microorganisms, the most sensitive *S. aureus*, *E. faecalis*, and *E. coli*. The results revealed MIC values of 0.0078% and 0.675% for OEO NPs and the free form, respectively, in *S. aureus*. 

Collectively, these findings show the versatility of systems obtained using polymeric NPs and their impact on antibacterial applications, also demonstrating that the EOs’ properties could be enhanced by nanotechnology.

### 5.4. Synthetic Biopolymers NPs

Synthetic macromolecules have been employed as nanocarriers for the controlled release of EOs. Examples of these biopolymers are Poly(DL-lactide-co-glycolide) (PLGA) and poly-ε-caprolactone (PCL).

Recently, it was reported that PLGA NPs loaded with EO from the *Laurus nobilis* L. plant, with an average size of 211.4 ± 4 nm, could maintain a sustained release during 72 h [138]. This property enhances the characteristics of EOs and decreases the needed doses to achieve adequate results. In this context, in 2022, it was reported that trans-cinnamaldehyde loaded in PLGA NPs increases its microbial effect 2-fold against *S. typhimurium* and *S. aureus*, presenting delivery after 72 h [116]. Also, in 2022, Artemisia vulgaris EO was entrapped in PLGA NPs, showing significant growth inhibitory effects against Gram-positive bacteria (*S. aureus* and *M. luteus*).

Regarding PCL, in 2021, eugenol was encapsulated in PCL and polyethyleneglycol NPs to evaluate their effect on both Gram-positive and Gram-negative bacteria [139]. The growth rate was analyzed after different times of exposure with 40 µM of eugenol-loaded PCL NPs and 40 µM of free eugenol, observing that the NPs presented a remarkable effect on the growth even at long time intervals derived from the long-term antibacterial efficiency of the nanocarriers. Similarly, s. *Thymus capitatus* and *Origanum vulgare* EOs were encapsulated in PCL nanocapsules, obtaining an average diameter of 171 and 175 nm and percentages of entrapment of 96 and 91, respectively. These systems and the pure EOs were evaluated in *S. aureus*, *E. coli*, *P. aeruginosa*, and *L. monocytogenes*. EOs-loaded NPs presented lower MIC and MBC values than the free EOs against *S. aureus*, *E. coli*, and *L. monocytogenes*, suggesting that their encapsulation in nanocarriers can increase the transport mechanisms and diffusion of the EOs across the bacterial cell membrane. For *P. aeruginosa*, no significant susceptibility to free EOs or EOs-loaded NPs was observed.

## 6. Antibiofilm Efficacy of EOs-Loaded Chitosan and Other Biopolymeric Nanoparticles

Although there is a wealth of information available regarding the antibiofilm properties of EOs and the successful encapsulation of EOs using different types of NPs, there is a notable lack of research on the utilization of chitosan and biopolymeric NPs loaded with EOs as an antibiofilm treatment. This section addresses this research gap by presenting the current state of knowledge through a comprehensive review of available research results and analyzing their potential implications.

### 6.1. Chitosan

Cinnamaldehyde was also encapsulated in chitosan NPs to evaluate the anti-quorum sensing of *Pseudomonas aeruginosa* [140]. The NPs presented an average size of 208 nm and an efficiency of entrapment of 65%. Moreover, the NPs reduced a 71.25% biofilm formation at sub-MIC concentrations (250 and 500 µg/mL). The authors also evaluated the motility of *P. aeruginosa* after NPs treatment because this property plays a crucial role in biofilm formation, development, and maturation. At 500 µg/mL, both cinnamaldehyde and NPs significantly affected the swimming and swarming motility of *P. aeruginosa*. Furthermore, the anti-quorum sensing was evaluated by light microscopic analysis (Figure 4), demonstrating that the treatment with NPs (500 µg/mL) presented a significant reduction in the biofilm thickness, compared with the cinnamaldehyde treatment and the untreated biofilm, suggesting its efficacy in attenuating the antibiotic-resistant biofilm formation.

A remarkable study on the inhibition of *S. mutants* biofilm by chitosan NPs was reported in 2019 [141]. The authors employed chitosan to encapsulate Mentha piperita EOs, an effective antimicrobial against Gram-positive and Gram-negative bacteria. The formulations presented an average size of 567.1 nm for empty NPs and 575.6 nm for the loaded NPs. The authors tested the effect of different concentrations of loaded NPs on the adherence of the bacterial cells to dental surfaces. The findings showed that the adherence was influenced in all treatment concentrations (7.5 to 250 µg/mL), suggesting that the formulation could inhibit the synthesis of adhesion molecules by *S. mutants*. Furthermore, the antibiofilm effect of nanoformulations was evaluated. The results showed that, even with the higher dose of free EOs (400 µg/mL), the biofilm formation inhibition was lower than observed with the lowest dose of EOs-loaded NPs (50 µg/mL).

In the same way, modified and unmodified chitosan was used as base material to entrap carvacrol in NPs [142]. Chitosan was modified by grafting 11-carbon (SH11) and 3-carbon (SH3) alkyl chains to obtain hydrophobic NPs and evaluate their antibiofilm properties in *P. aeruginosa* biofilms. As mentioned above, carvacrol is a monoterpenoid phenol extracted from oregano that presents antimicrobial activity against Gram-positive and Gram-negative bacteria. The formulations showed a carvacrol entrapment efficiency of 50% for unmodified chitosan NPs, 25.1% for chitosan–SH3 NPs, and 68.8% for chitosan– SH11 NPs, and all of them exhibited the ability to eradicate biofilms, even the carvacrol-free NPs. However, a higher percentage of biofilm biomass disruption was observed for the carvacrol–chitosan–SH11 NPs, which reduced approximately 54% on the preformed *P. aeruginosa* biofilm. The penetration of NPs was analyzed by confocal microscopy (Figure 5). There was less fluorescence intensity in the biofilms treated with chitosan NPs than those treated with chitosan–SH11, suggesting a better distribution inside the biofilm.

### 6.2. Synthetic Macromolecules NPs 

PLGA has been employed as a carrier for molecules derived from EOs, such as carvacrol, cinnamaldehyde, and farnesol, to inhibit biofilm formation [143,144]. For example, through the solvent displacement method, Ianitelli et al. [145] formulated OEO-loaded NPs using PLGA, a biocompatible polymer widely used for drug delivery systems. An antibiofilm assay was performed using the strain *S. epidermidis*, and rheological tests of biofilm with carvacrol-loaded NPs and free form, producing a considerable reduction in their steady-flow viscosity, obtaining values of 5.5 × 10^5^, 6.9 × 10^5^, 1.1 × 10^6^ Pa s, respectively. The results exhibit the enhancement in antibiofilm properties of OEO after nanoencapsulation, affecting the mechanical stability of biofilm and inhibiting development.

Similarly, Sebelemetja et al. entrapped a subfraction (F5.1) of the crude extract of Dodonaea viscosa var. angustifolia in PLGA/PEG/PVA NPs through the double emulsion method [146]. The antimicrobial properties of these NPs were evaluated in *Streptococcus mutans* cultures, the most relevant biofilm-forming bacterium inhabiting the mouth cavity and associated with dental plaque. Based on the MIC and minimal bactericidal concentrations obtained in some experiments, the authors chose subinhibitory concentrations of F5.1, F5.1/NPs, and NPs to determine their effect on biofilm formation in *S. mutans*. Their findings demonstrated that bacterial counts in the biofilm decreased in the presence of F5.1 and F5.1/NPs compared with the control. Subfraction F5.1 inhibited biofilm formation at the concentration of 0.1 mg/mL and F5.1/NPs at 0.2 mg/mL; however, due to the entrapment efficiency (44%), the inhibition of biofilm using NPs was achieved employing just 0.88 mg/mL of F5.1 subfraction.

Equally, Gursu et al. evaluated the potential of cinnamaldehyde-loaded PLGA NPs as pre- and post-treatment against Candida albicans biofilm [144]. The loaded NPs had an average size of 136.7 nm and presented an MIC value of 250 µL/mL. In contrast, cinnamaldehyde exhibited an MIC of 32.7 µL/mL. Although the MIC of cinnamaldehyde NPs was higher than free cinnamaldehyde, it is known that the poor solubility and side effects of the free EOs (such as irritability) impede their application as treatments. Furthermore, the anti-quorum sensing activity was evaluated based on microtoxicity measurement device data, showing that the cinnamaldehyde-loaded PLGA NPs were three times more toxic than cinnamaldehyde alone.

The efficacy of oregano and thyme loaded in poly(-caprolactone) nanocapsules was evaluated against four environmental microorganisms: *Pleurotus eryngii*, *Purpureocillium lilacinum*, *Pseudomonas vancouverensis*, and *Flavobacterium* sp. [147]. The authors reported MICs of 0.125 mg/mL and 0.5 mg/mL for fungal and bacterial strains, respectively.

The synergy of EOs and different antibiotics/drugs in nanoformulations has also been evaluated in different biofilms. In this context, the combination of thymol-loaded NPs and amikacin antibiotics was evaluated in *Salmonella enterica serovars*, which causes typhoidal and nontyphoidal salmonellosis [148]. The authors reported that the minimum biofilm inhibitory concentration was reduced four to eight times with the synergistic mechanism compared with the inhibition of amikacin alone. 

## 7. Other Nanostructured Systems for EOs Delivery

Several notable studies have demonstrated the efficacy of different nanoformulations for encapsulating EOs, thereby preserving and enhancing their antimicrobial properties. The following works highlight outstanding results in this area:

In 2022, Elghobashy et al. [149] conducted a study where they developed a chitosan/alginate nanocomposite loaded with thyme and garlic EOs. The researchers fabricated NPs using this composite and loaded them with the EOs, resulting in particles ranging from 150 to 200 nm in size and possessing a Z-potential of approximately 30 mV. The thyme-oil-loaded composites exhibited enhanced inhibitory effects against various bacteria, including *S. aureus*, *E. coli*, and *A. hydrophila*, with MICs of 10, 15, and 12.5 μgmL^−1^, respectively. These MIC values were comparable to ampicillin, a commonly used antibiotic. Interestingly, when the composites were loaded with a combination of both EOs, they demonstrated an increased antibacterial capacity, specifically against Gram-positive bacteria. Notably, the antibacterial efficacy of the composites far surpassed that of the pure oils alone, suggesting a synergistic effect between the properties of the oils and the nanoscale structure of the composite. The nanometer scale is believed to be a critical factor for facilitating cell membrane penetration, contributing to the heightened antibacterial activity observed in the study.

In 2021, Ullah et al. developed a novel wound dressing utilizing cellulose acetate nanofibers loaded with sambong oil, marking the first instance of incorporating this EO into a nanosystem [150]. The researchers investigated the antibacterial efficacy of these dressings against *E. coli* and *S. aureus*, two strains known to impede the wound healing process. By conducting a qualitative diffusion assay test, they observed an increase in the size of the inhibition zone as the concentration of the oil in the dressings increased, ranging from 1 to 4 mm. Additionally, the researchers demonstrated a reduction in the growth of microorganisms using the method of serial dilutions, indicating an approximately 20% decrease. Importantly, this reduction percentage depended on the duration of contact between the dressing and the bacteria, highlighting the importance of the interaction time. These findings indicate the potential of this nanosystem as an effective means of combating bacterial infections in the context of wound healing.

Conversely, a coating based on an alginate emulsion containing myrtle oil showed inhibitory effects on *L. monocytogenes* in kasar cheese, according to the findings of Yemis et al. [151]. The antimicrobial properties were evaluated through a serial dilution method in a microplate, and significant differences were observed when comparing the nanoemulsion (with particle sizes ranging between 122.7 and 157 nm) to the emulsion (with particle sizes ranging between 1184 and 1490 nm). The MIC was reduced by more than 30% when utilizing nanoemulsion, highlighting the crucial role of size reduction in enhancing antibacterial activity. Furthermore, the researchers assessed the antimicrobial activity of the coating against the same microorganism in kasar cheese, obtaining favorable results. These findings suggest the potential application of the myrtle-oil-based coating as a functional protective layer for inhibiting *L. monocytogenes* in cheese.

In 2020, Beikzadeh et al. developed a highly effective material composed of cellulose acetate nanofibers encapsulating lemon myrtle EO, which exhibited potent antimicrobial activity [152]. The efficacy of these materials was evaluated against *E. coli* and *S. aureus* strains using a modified version of the standardized Japanese industrial method [153]. This method involves inoculating the fibers with a known quantity of CFU and subsequently assessing the surviving bacteria after a specified contact time. The study results demonstrated that the materials possessed robust antibacterial activity, eliminating 100% of the bacterial population in samples with a high concentration of EOs. Even in samples with a minimal oil content of 2%, a significant reduction of at least two orders of magnitude in the bacterial count was observed. Notably, the study also showcased the efficiency of the nanosystems in preserving EOs. The cellulose acetate nanofibers successfully maintained their antimicrobial properties even after a storage period of two months, emphasizing their stability and long-term functionality.

## 8. Conclusions and Future Perspectives

The emergence of bacterial resistance to antibiotics has reached a critical point in recent decades, resulting in numerous deaths caused by infections that no longer respond to conventional medical treatments. Additionally, discovering new forms of bacterial adaptation to existing antibiotics paints a grim picture. Furthermore, bacterial biofilms pose significant challenges in combating microorganisms, making the development of new treatments more complex, adding that infections often involve multiple bacterial species with varying resistance levels due to their adaptation to different surfaces and poorly controlled microenvironments.

Although some approaches have been explored, cost-effective alternatives are still lacking. In this context, EOs offer promise as an alternative to antibiotics because they are less likely to be detected by bacteria. However, using carriers is necessary to facilitate their penetration into biofilms and avoid some side effects, such as irritation in topical application.

Macromolecule-based NPs, with their versatility, hold the potential as an option to address this global health problem. Chitosan has gained popularity as the most extensively studied biopolymer for this purpose due to its unique properties and characteristics, showing exciting results in biofilm eradication, as this review presented.

However, there is a need for new approaches utilizing different materials for delivering antimicrobial molecules and achieving enhanced biofilm internalization, inhibition, and disaggregation given biofilms’ intricate composition and dynamic nature. In this context, synthetic biopolymers could be an adequate option due to their excellent results in other nanotechnological applications and their release profiles, which allow the effects of oils and phytochemicals to endure much longer.

Also, future research may explore new strategies, such as surface modification using molecules that could disaggregate the biofilm in early or mature stages. Interesting findings demonstrated that antimicrobial peptides, considered the next generation of antibiotics, cause physical damage in microorganisms, decreasing the possibility of developing resistance [18,154]. In the same way, these peptides helped in the biofilm disruption, making them excellent candidates to be employed as functional ligands on NPs’ surface, to act in a synergic manner with other biomolecules.

Additionally, we believe that future approaches should extend the cultivation times of microbial biofilms because mature biofilms are more complex and robust than young biofilms. Also, the analysis of the interaction of NPs and biofilms should be deeper and longer than the current reports to find some relations between the biofilm composition and nanosystems.

To summarize, biopolymeric NPs represent a promising strategy for several biomedical fields and the use of EOs; however, despite the vast potential of these technologies, their clinical application in bacterial resistance has not been entirely achieved. An explanation for this problem is the lack of intensive analysis of the results of the hard work combatting the biofilms. Furthermore, although some interactions between NPs and biofilms have been elucidated, further studies are required in real-world environments to understand these interactions’ complexities. In this context, new developments will need to be toughly tested in vivo, and continued research is necessary to achieve these advancements.

## Figures and Tables

**Figure 1 ijms-25-00082-f001:**
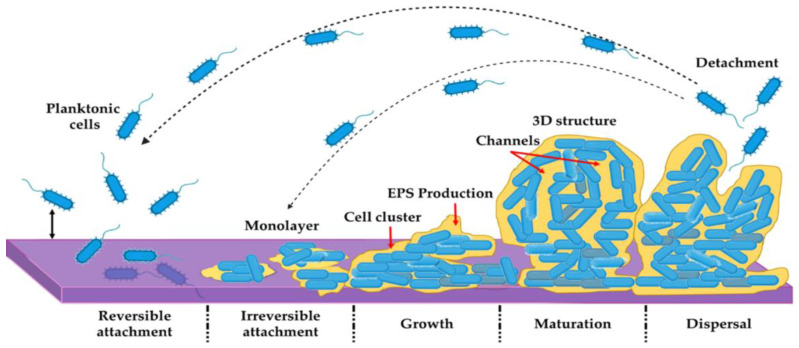
Schematic representation of a biofilm formation (see detailed explanation in text).

**Figure 2 ijms-25-00082-f002:**
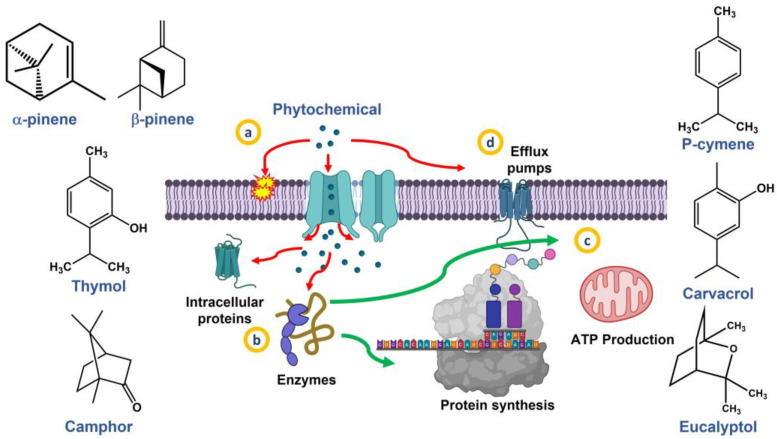
Chemical structure of some phytochemicals present in EOs (rosemary, oregano, and clove) and their main antimicrobial mechanisms. Image created with Biorender.com.

**Figure 3 ijms-25-00082-f003:**
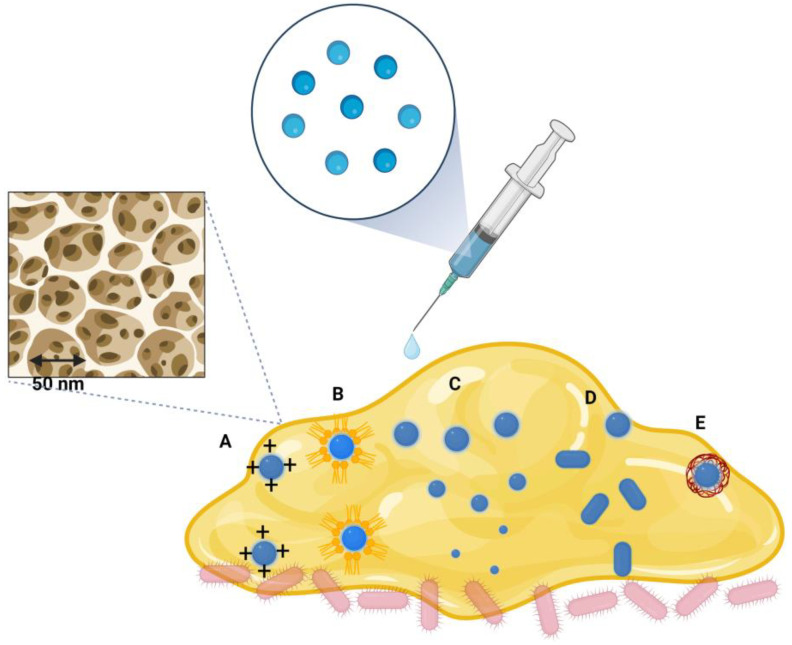
Mechanisms of interaction of NPs with the biofilm (see details in the text). (A) The positive electrical charge of the NPs favors penetration, while the negative charges delay diffusion. (B) NPs with hydrophobic surfaces present greater interaction with bacteria. (C) NPs with a smaller size show greater diffusion. (D) The rod-like shape exhibits greater penetration capacity than the spherical ones. (E) The formation of the biomolecular corona can retain the transit of the NP on the surface of the biofilm by an increase in size and a decrease in the self-diffusion coefficient. The upper left represents one of the approximate pore diameters of the open areas of the biofilms.

**Figure 4 ijms-25-00082-f004:**
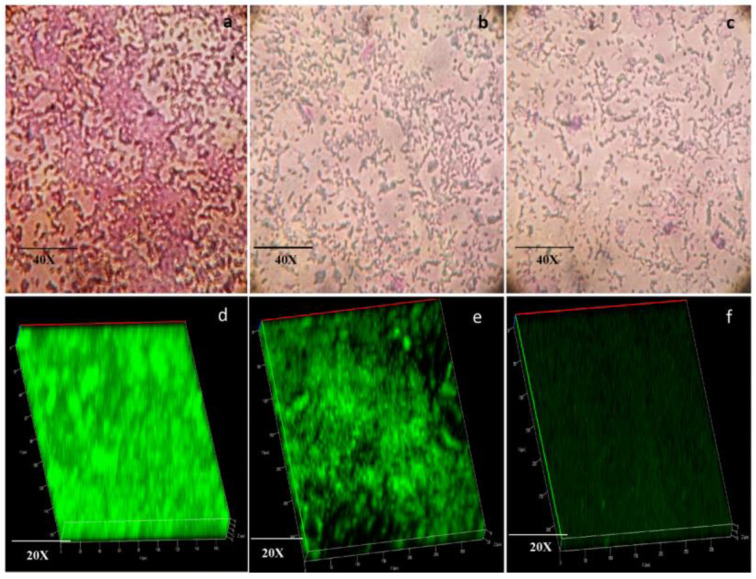
Microscopic analysis of *P. aeruginosa* PAO1 biofilm on treatment with sub-MIC concentration (500 µg/mL) of cinnamaldehyde and cinnamaldehyde-loaded NPs compared to untreated control [140]. (**a**) Light microscopic observation of biofilm of untreated control; (**b**) light microscopic observation of biofilm treated with cinnamaldehyde; (**c**) light microscopic observation of biofilm treated with NPs; Confocal Laser Scanning Microscopy image of (**d**) biofilm of untreated control; (**e**) biofilm treated with cinnamaldehyde; (**f**) biofilm treated with cinnamaldehyde-loaded NPs.

**Figure 5 ijms-25-00082-f005:**
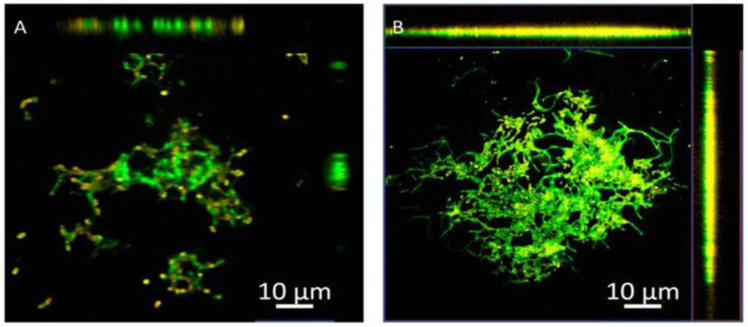
Representative 3D projection of image Z-stacks showing the distribution of bacterial cells (green) in *P. aeruginosa* biofilms and chitosan nanoparticles (orange): (**A**) chitosan; (**B**) chitosan–SH11 [142].

**Table 1 ijms-25-00082-t001:** Recent publications of EOs-based polymeric nanosystems.

Biopolymer	Essential Oil	Method	Size (nm)	Potential Z (mV)	PDI	Application	Targeted Bacteria	Highlights	Ref.
Chitosan	Clove (*Eugenia caryophyllata*)	Ionic gelation	223–44.5	+20.14 t +10.114	0.117–0.337	Antioxidant and antibacterial	*Staphylococcus aureus*, *Listeria monocytogenes*, *Escherichia coli*, *Salmonella typhi*	The entrapment of the EO ranged from 55.8 to 63.1%. The inhibition halo improved compared to the free form of clove essential oil.	[103]
	Clove (*Eugenia caryophyllata*)	Emulsion–ionic gelation	268.47	+22.45	n.r.	Antifungal	*Aspergillus niger*	The encapsulation of clove essential oil enhanced the antifungal index from 65% to 100%.	[104]
	Cinnamon (*Cinnamomum zeylanicum*)	Ionic gelation	215–527	n.r.	0.357–0.617	Antibacterial	*Pseudomonas aeruginosa*	The application of cinnamon essential oil nanoparticles in chilled pork decreased the microbial growth from 7.13 lg cfu g^−1^ to 5.15 lg cfu g^−1^ after 15 days.	[105]
	Cinnamon (*Cinnamomum zeylanicum*) and Thyme (*Thymus capitatus*)	Ionic gelation	29.3	+22.9	0.3	Antibacterial	*Staphylococcus aureus* *Enterococcus* sp., *Escherichia coli*, *Klebsiella pneumoniae* and *Pseudomona aeruginosa*	Cinnamon essential oil nanoparticles presented a diminution of the percentage of proliferation around 23–54% compared to the free form.	[106]
	*Homalomena pinedora*	Ionic gelation	70	+24.1	0.176	Antimicrobial activity broad-spectrum against diabetic wound pathogens	*Bacillus cereurs*, *Staphylococcus aureus*, *Escheria coli*, *Proteus mirabilis*, *Klebsiella pneumoniae*, and *Pseudomonas aeruginosa*	The nanoparticle also showed a concentration-dependent behavior on time-kill assay. The nanoparticles in the 3D collagen wound models reduced microbial growth by 60–80%.	[107]
	Nettle (*Urtica dioica* L.)	Emulsion–ionic gelation in two stages	208–369	+14 to +30	0.153–0.412	Antibacterial activity	*Staphylococcus aureus*, *Bacillus cereus*, *Listeria monocytogenes*, *Escherichia coli*, and *Salmonella typhi*	Nettle-loaded chitosan NPs presented the highest antibacterial inhibitory activity, showing an increment in the antimicrobial properties of the free nettle.	[108]
	Aegle marmelos	Ionic gelation	n.r.	n.r.	n.r.	Antibacterial activity	*Carbapenem-resistant K. pneumoniae*	The minimum inhibitory activity of AMEOs-CHs NPs against CR K. pneumoniae was observed at 40 μg/mL. In addition, the results confirmed that the AMEOs-CHs have carbapenemase enzyme degradation ability. DNA fragmentation and suppression of VIM 1 and IMP 1 gene expression indicated that the AMEOs-CHs were excellent antibacterial material.	[109]
Alginate	Oregano (*Origanum vulgare* L.)	Emulsification and electrostatic gelation	320	−25	n.r.	Antibacterial	*Staphylococcus aureus*, *Streptococcus pyogenes*, *Enterococcus faecalis*, *Escherichia coli*, *Pseudomonas aeruginosa*, and *Yersinia enterocolitica*	The minimal inhibitory concentrations of OEO-NP on a panel of Gram-positive and Gram-negative pathogens are 4–32-fold lower than those of OEO. OEO-NP inhibited the respiratory activity of the bacteria to a lower extent than free OEO; however, the minimal bactericidal concentrations remain significantly lower.	[16]
Zein	Cinnamon essential oil	Antisolvent method/Emulsion	660	31.23	0.271	Antimicrobial activity	*Alternaria alternate* and *Botrytis cinerea*	The formulation showed superior antibacterial performance than pure essential oil.	[110]
	Thymol	Antisolvent method	177.5–240.2	−39.6 to −37.8	0.27	Antimicrobial activity	*Staphylococcus aureus*	Thymol loadings provided the nanoparticles antimicrobial activity against tested bacteria on agar diffusion assay and DPPH radical scavenging activity in a dose-dependent manner.	[111]
Cellulose	Peppermint, cinnamon, and lemongrass	Nanoprecipitation	150–200	−42 to −38	n.r.	Antimicrobial activity	*Staphylococcus aureus*, *Pseudomonas aeruginosa*, *Escherichia coli*	Cinnamon-EOs-encapsulated NCs presented significant growth inhibition of all bacterial strains, especially *E. coli*; peppermint-EOs-encapsulated NCs demonstrated a low inhibitory activity against *S. aureus* and *C. albicans*. Meanwhile, lemongrass-EOs-encapsulated NCs slightly inhibited the development of *E. coli*. *P. aeruginosa* strain exhibited the highest resistance to the tested NCs.	[112]
PCL	Carvacrol (*Thymus capitatus*) and thymol (*Origanum vulgare*)	Nanoprecipitation	200	−10 to −11	0.05–0.09	Antibacterial, antifungal, and antibiofilm activities	*Staphylococcus aureus*, *Escherichia coli*, and *Candida albicans*	Th-NCs and Or-NCs were more effective against all tested strains than pure EOs and at the same time were not cytotoxic on HaCaT (T0020001) human keratinocyte cell line.	[113]
	Tea tree	Interfacial deposition	268	+31	0.204	Topical acne treatment	*Cutibactrium acnes*	The nanosystem showed significant anti-C. acnes activity, with a 4× reduction in the minimum inhibitory concentration, compared to TTEO and a decrease in C. acnes cell viability, with an increase in the percentage of dead cells (17%) compared to control (6.6%) and TTEO (9.7%).	[114]
	Palmarosa (*Cymbopogon martini Roxb*)	Nanoprecipitation	282.1	−27.2	0.116	Antioxidant and antimicrobial	*Escherichia coli* and *Staphylococcus aureus*	The nanoparticles inhibited the strains; however, at higher concentrations of E. coli, there was no inhibition but showed sensitivity. This lower sensitivity to Gram-negative bacteria compared to the free compounds can be explained by the lipopolysaccharide outer layer in their membrane, limiting the diffusion of these hydrophobic compounds.	[115]
PLGA	Trans-cinnamaldehyde	Emulsion-evaporation	277.3–295	+9.54	0.16	Antimicrobial	*Salmonella Typhimurium* and *Staphylococcus aureus*	The antimicrobial activity of EO-NPs was lower than the free EO form. However, PLGA nanoparticles showed different release profiles depending on environmental pH and chitosan presence, demonstrating a potential pH-triggered mechanism for natural antimicrobial compound release.	[116]
	Fennel	Emulsion-diffusion	123.19	+23	0.051	Antibacterial activity	*Staphylococcus aureus*	The antimicrobial activity of EO-NPs (MIC: 3 µg/mL) was much higher than that of the pure EO (MIC: 12.5 µg/mL).	[117]
	*Pistacia lentiscus* L. var. chia	Solvent evaporation	239.9 and 286.1	−29.1 and −34.5	0.081 and 0.167	Treatment of minor skin inflammations	*Escherichia coli* and *Bacillus subtilis*	PLGA/PVA NPs demonstrated increased stability over time and more sustained release compared to PLA/LEC NPs. The antimicrobial activity studies confirmed the activity of the essential oil against *E. coli* and *B. subtilis*. At the same time, no such results were revealed from an analogous survey conducted in NPs, probably due to the low concentration of EO at specific time intervals.	[118]
	Garlic (*Allium sativum* L.)	Emulsion/ solvent evaporation	201–319	−36.69 to −35.37	0.1–0.36	Antibacterial activity	*Escherichia coli* and *Staphylococcus aureus*	The antibacterial activities show enhancement by 70–78% of bacterial inhibition compared with a GO bulk solution.	[18]

PLGA: Poly(DL-lactide-co-glycolide), PCL: Poly(ε-caprolactone), n.r.: not reported, PVA: poly(vinyl alcohol), LEC: Lecithin.

## Data Availability

Not applicable.

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
