# Peer review of "Essential-Oils-Loaded Biopolymeric Nanoparticles as Strategies for Microbial and Biofilm Control: A Current Status"

_ijms, 2023, doi:10.3390/ijms25010082_

Round 1

Reviewer 1 Report

Comments and Suggestions for Authors

The review is well-structured and provides a general overview about the subject of biopolymeric nanoparticles loaded with essential oils as antimicrobials and film growth control.

In paragraph 3, the topic about essential oils and their use as antimicrobial and anti-biofilm agents should be better introduced. Why only cinnamon, rosemary, clove and oregano oils are described for the antimicrobial and anti-biofilm properties?

Why among all natural biopolymers the attention of the review is only about chitosan, zein and alginate as antimicrobial strategy (paragraph 5) and only about chitosan as anti-biofilm strategy (paragraph 6)?

Paragraph Synthetic macromolecule NPs is incorrectly referred as 5.2 instead of 6.2.

Paragraph 5.4 and Paragraph 5.2 (alias 6.2) can be named both as “Synthetic biopolymer NPs”.

Author Response

The review is well-structured and provides a general overview about the subject of biopolymeric nanoparticles loaded with essential oils as antimicrobials and film growth control.

In paragraph 3, the topic about essential oils and their use as antimicrobial and anti-biofilm agents should be better introduced.

We agreed with this commentary. Following all the observations the introduction was resumed to remark on the main objective, which is related to the EOs and nanotechnology employed in bacterial resistance. For this reason, in the third paragraph information about EOs is found.

Why only cinnamon, rosemary, clove and oregano oils are described for the antimicrobial and anti-biofilm properties?

We understand this concern. We emphasized the information on those essential oils due to their antibiofilm capacity. Meanwhile, phytochemicals, in general, have been widely studied regarding their antimicrobial activity, and over 100,000 different phytochemical types have been described, these molecules present interesting behavior talking about combating biofilm. Particularly, the components in cinnamon and oregano oil make them two of the most promissory alternative treatments to inhibit or combat biofilm and its complications. Furthermore, other reviews analyzed the information about more EOs and their relation with microbial resistant, but not in the anti-biofilm capacity of them.

Why among all natural biopolymers the attention of the review is only about chitosan, zein and alginate as antimicrobial strategy (paragraph 5) and only about chitosan as anti-biofilm strategy (paragraph 6)?

This is a very important observation. In this review, we attempt to analyze primarily the effect of natural biopolymers. Meanwhile, there are many examples of these materials in nature, mainly chitosan, zein, and alginate have been studied as nanoparticles to attack microorganisms.

In this review, we wanted to focus on natural materials that interact with microorganisms, which could somehow increase the action of nanotechnological developments. For example, chitosan has been studied in depth due to the properties derived from its surface charge, making it one of the candidate polymers for working nanosystems. On the other hand, alginate has been analyzed because it is a biocompatible and economically affordable material. Zein, for its part, is a natural material that in recent years has received the attention of scientists due to its qualities. Knowing that there are many natural options that could be analyzed in the review, these polymers were chosen because they are the most studied in these applications. In addition, we also wanted to give the reader a global experience of the current situation of polymeric nanosystems developed to combat antibiotic resistance, consequently, we added the synthetic polymers section.

Paragraph Synthetic macromolecule NPs is incorrectly referred as 5.2 instead of 6.2.

We are so sorry about this mistake; we corrected the section title.

Paragraph 5.4 and Paragraph 5.2 (alias 6.2) can be named both as “Synthetic biopolymer NPs”.

Thank the reviewer for this timely observation.  The titles were changed and now the manuscript presents coherent sections.

Reviewer 2 Report

Comments and Suggestions for Authors

Thank you for submitting the manuscript "Essential oils-loaded biopolymeric nanoparticles as strategies for microbial and biofilm control: A current status" to International Journal of Molecular Science. Overall, the topic deserves a review manuscript because even though it is a new subject, there are already several research articles reporting results. However, although the subject is interesting, the general part of the manuscript is almost half the size of the manuscript (until item 4) and I believe that this part could be reduced to actually enhance the part of the manuscript intended for NP of EO.

Another general issue relating to the entire manuscript is that all scientific names need to be checked as there are several without italic formatting.

Consider citing https://doi.org/10.1016/B978-0-12-823791-5.00009-0 and https://doi.org/10.1039/9781839168048-00101.

Line#18: activities

Line#:99-106: although you described that talking about NP is one of the objectives of the review article, here it seems to be a secondary objective. This does not appear to be true since it constitutes the title of the manuscript. Therefore, this part needs to be reformulated to make the NP part of the manuscript stand out or the title needs to be changed.

Comments on the Quality of English Language

 Minor editing of English language required.

Author Response

Thank you for submitting the manuscript "Essential oils-loaded biopolymeric nanoparticles as strategies for microbial and biofilm control: A current status" to International Journal of Molecular Science. Overall, the topic deserves a review manuscript because even though it is a new subject, there are already several research articles reporting results. However, although the subject is interesting, the general part of the manuscript is almost half the size of the manuscript (until item 4) and I believe that this part could be reduced to actually enhance the part of the manuscript intended for NP of EO.

We agree with the reviewer's observation. The manuscript was resumed in order to be more specific and achieve the main objective.

Another general issue relating to the entire manuscript is that all scientific names need to be checked as there are several without italic formatting.

Thank you for this pertinent correction. The manuscript was modified.

Consider citing https://doi.org/10.1016/B978-0-12-823791-5.00009-0 and https://doi.org/10.1039/9781839168048-00101.

We thank the reviewer for this suggestion. This reference was added.

Line#18: activities

Line#:99-106: although you described that talking about NP is one of the objectives of the review article, here it seems to be a secondary objective. This does not appear to be true since it constitutes the title of the manuscript. Therefore, this part needs to be reformulated to make the NP part of the manuscript stand out or the title needs to be changed.

Following this timely commentary, we reformulated and added information about NPs to be coherent with the manuscript's aim.